# Repurposing Clemastine to Target Glioblastoma Cell Stemness

**DOI:** 10.3390/cancers15184619

**Published:** 2023-09-18

**Authors:** Michael A. Sun, Rui Yang, Heng Liu, Wenzhe Wang, Xiao Song, Bo Hu, Nathan Reynolds, Kristen Roso, Lee H. Chen, Paula K. Greer, Stephen T. Keir, Roger E. McLendon, Shi-Yuan Cheng, Darell D. Bigner, David M. Ashley, Christopher J. Pirozzi, Yiping He

**Affiliations:** 1The Preston Robert Tisch Brain Tumor Center, Duke University Medical Center, Durham, NC 27710, USA; michael.a.sun@duke.edu (M.A.S.); rui.yang@duke.edu (R.Y.); heng.liu@duke.edu (H.L.); wenzhe.wang@duke.edu (W.W.); nathan.reynolds479@duke.edu (N.R.); kristen.jckamp@gmail.com (K.R.); oredgreen@gmail.com (L.H.C.); paula.greer@duke.edu (P.K.G.); stephen.keir@duke.edu (S.T.K.); roger.mclendon@duke.edu (R.E.M.); darell.bigner@duke.edu (D.D.B.); david.ashley@duke.edu (D.M.A.); 2Department of Pathology, Duke University Medical Center, Durham, NC 27710, USA; 3Pathology Graduate Program, Duke University Medical Center, Durham, NC 27710, USA; 4The Ken & Ruth Davee Department of Neurology, Lou and Jean Malnati Brain Tumor Institute, The Robert H. Lurie Comprehensive Cancer Center, Simpson Querrey Institute for Epigenetics, Northwestern University Feinberg School of Medicine, Chicago, IL 60611, USA; xiao.song@northwestern.edu (X.S.); bo.hu@northwestern.edu (B.H.); shiyuan.cheng@northwestern.edu (S.-Y.C.); 5Department of Neurosurgery, Duke University Medical Center, Durham, NC 27710, USA

**Keywords:** clemastine, glioblastoma, stemness, Emopamil Binding Protein (EBP)

## Abstract

**Simple Summary:**

Brain tumor-initiating cells (BTICs) drive tumor progression and resistance to treatments, posing formidable challenges to advancing effective treatments against glioblastoma (GBM). We postulated that inducing BTIC differentiation can serve as a solution to diminishing their malignant features. In this study, we found that clemastine, an over-the-counter oral medication for allergy relief, attenuated the propagation and promoted the differentiation of BTICs, and we uncovered the indispensable role of EBP (Emopamil-binding protein) in maintaining the BTIC population. Taken together, our study implicates specific pathways in the perpetuation of BTICs, and identifies a non-oncology drug with a well-established safety profile that can be repurposed to mitigate the malignant properties of BTICs in GBM.

**Abstract:**

Brain tumor-initiating cells (BTICs) and tumor cell plasticity promote glioblastoma (GBM) progression. Here, we demonstrate that clemastine, an over-the-counter drug for treating hay fever and allergy symptoms, effectively attenuated the stemness and suppressed the propagation of primary BTIC cultures bearing *PDGFRA* amplification. These effects on BTICs were accompanied by altered gene expression profiling indicative of their more differentiated states, resonating with the activity of clemastine in promoting the differentiation of normal oligodendrocyte progenitor cells (OPCs) into mature oligodendrocytes. Functional assays for pharmacological targets of clemastine revealed that the Emopamil Binding Protein (EBP), an enzyme in the cholesterol biosynthesis pathway, is essential for BTIC propagation and a target that mediates the suppressive effects of clemastine. Finally, we showed that a neural stem cell-derived mouse glioma model displaying predominantly proneural features was similarly susceptible to clemastine treatment. Collectively, these results identify pathways essential for maintaining the stemness and progenitor features of GBMs, uncover BTIC dependency on EBP, and suggest that non-oncology, low-toxicity drugs with OPC differentiation-promoting activity can be repurposed to target GBM stemness and aid in their treatment.

## 1. Introduction

Glioblastoma (GBM) tumors demonstrate striking aggressiveness and therapeutic resistance, which are driven by brain tumor-initiating cells (BTICs), a subpopulation of GBM cells that exhibit phenotypic plasticity and stem/progenitor-like features [1,2,3,4,5,6]. The pathological ramifications of GBM plasticity and stemness are highlighted by the heterogeneous nature of GBM with the presence of various subtypes, including proneural (PN), classical (CL), and mesenchymal (MES) subtypes [7,8,9,10,11,12]. Among the GBM subtypes, PN tumors are characterized by stem/progenitor-like signatures, a failure to respond to more aggressive chemo- and radio-therapy [9,13], and the ability to undergo the process of proneural-mesenchymal transition (PMT) [14,15]. Facilitated and driven by GBM plasticity and stemness, these tumors can transition from one subtype to another in the course of tumor progression and recurrence, and give rise to the most aggressive tumors with the worst prognosis, the MES subtype GBMs [9,13]. In addition, a previous study has provided evidence to suggest that the PN-like precursor cells can serve broadly as progenitors for GBMs and give rise to various subtypes of tumor cells [16]. Collectively, these findings support targeting GBM cell stemness as a strategy for mitigating GBM progression.

Numerous studies have provided critical insights into the mechanism of stemness maintenance in GBM [17,18,19,20,21], and offered promising strategies, including epigenetic and metabolic approaches, for targeting this property of GBM cells [22,23,24,25,26,27]. One alternative strategy for targeting stemness and plasticity of cancer cells is to promote their differentiation, an approach that shows great promise in blood cancers [28]. Although employing this strategy for solid tumors has proven more complicated, recent studies have supported the feasibility of attenuating tumor cells’ stemness/plasticity features and mitigating tumor progression in solid tumors via inducing their differentiation [29,30,31]. In gliomas, IDH1 mutant-induced differentiation blockage has been exploited for therapeutic purposes [32,33], and metabolism-based strategies for directing GBM differentiation have been proposed [34,35]. In addition to the benefit of attenuating stemness, plasticity, and self-renewal capacity, differentiation of tumor cells in solid tumors, including gliomas, can potentially promote tumor cell senescence and sensitize gliomas to immunotherapy, suggesting its multifaceted benefits from a therapeutic perspective [36,37,38,39].

Therapeutic differentiation of oligodendrocyte progenitor cells (OPCs) to become myelinating oligodendrocytes has been extensively pursued for treating multiple sclerosis (MS), a demyelinating disease that affects millions of patients worldwide. Several drugs are shown to effectively differentiate OPCs to myelinating oligodendrocytes in various in vivo disease models [40,41,42]. Notably, the PN subtype GBMs display molecular signatures reminiscent of OPCs, both at the individual tumor level and at the single cell level, suggesting their OPC origin and progenitor features [9,12]. We hypothesized that diminishing the progenitor features of these GBM cells via induction of differentiation can suppress their stemness and tumorigenicity. We identified clemastine, a low-toxicity, non-oncology drug for alleviating allergy symptoms with high potency in promoting OPC differentiation [40,41,42], as an agent that can be used to inhibit the propagation of BTIC cultures. Using BTIC cultures bearing *PDGFRA* amplification to represent PN subtype GBMs, we found that the tumor suppressive effects of clemastine were accompanied by diminished stemness of tumor cells and altered molecular signatures indicative of more differentiated states. Corroborating these findings, loss-of-function assays of putative pharmacological targets of clemastine identified the Emopamil Binding Protein (EBP), a crucial enzyme in the cholesterol biosynthesis pathway, as an essential protein for maintaining BTIC proliferation and stemness. Finally, clemastine suppressed the in vivo tumorigenicity of a mouse glioma cell line resembling the PN subtype of GBMs. This study identifies new candidate proteins in GBM cells that can be therapeutically targeted (i.e., EBP), and suggests that clemastine and other agents capable of inducing OPC differentiation can be leveraged for attenuating the stemness/plasticity of GBM cells, particularly those bearing PN features. Our findings support the principle of repurposing non-oncology drugs for inducing differentiation and thus mitigating GBM stemness.

## 2. Materials and Methods

### 2.1. Cell Lines

Primary GBM cultures (BTIC#102, BTIC#148, BTIC#127, and BTIC#095) were derived with consent from tumor tissues of patients at the Duke Brain Tumor Center and cultured as previously described [43,44]. GBM cultures GSC-1123 and JK-046 were derived and cultured as previously described [45,46]. Briefly, the GBM cell lines were maintained as BTICs in human neural stem cell (NSC) media containing NeuroCult NS-A basal medium (human, STEMCELL Technologies, #05750), supplemented with NeuroCult NS-A proliferation kit (human, STEMCELL Technologies, #05751, Vancouver, BC, Canada), human recombinant EGF (20 ng/mL, STEMCELL Technologies, #78006.2, Vancouver, BC, Canada), human recombinant FGF (10 ng/mL, STEMCELL, #78134.1), and heparin sodium salt (2 µg/mL, MilliporeSigma, #H3149-100KU, Darmstadt, Germany). The mouse glioma cell line (C266-6-IC-12/12, simplified as “C266”), established from tumors originating from *Trp53*^−/−^ mouse NSC transduced with retrovirus-*Pdgfb*, was cultured in standard mouse NSC media containing NeuroCult basal medium (mouse & rat, STEMCELL Technologies, #05700, Vancouver, BC, Canada), supplemented with a NeuroCult proliferation kit (mouse & rat, STEMCELL Technologies, #05702, Vancouver, BC, Canada), human recombinant EGF (20 ng/mL), human recombinant FGF (10 ng/mL), and heparin sodium salt (2 µg/mL). Normal human astrocytes (Lonza, #CL-2693-FV, Basel, Switzerland) were purchased from the Duke Cell Culture Facility and cultured in growth media containing MCDB 131 medium (no glutamine, Thermo Fisher Scientific, #10372019, Waltham, MA, USA), supplemented with ascorbic acid (75 µg/mL, Lonza, #CC-4398, Basel, Switzerland), human recombinant insulin (20 µg/mL, Lonza, #BE03-033E20, Basel, Switzerland), human recombinant EGF (2 ng/mL), antibiotic-antimycotic (1×, Thermo Fisher Scientific, #15240062, Waltham, MA, USA), fetal bovine serum (3%, Cytiva, #SH30071.03T, Marlborough, MA, USA), and GlutaMAX supplement (2 mM, Thermo Fisher Scientific, #35050061, Waltham, MA, USA). Cells were grown as suspension or on laminin-coated plates at 37 °C and 5% CO_2_, and experiments were performed within 30 passages after cells were thawed.

shRNA-mediated gene knockdown cells were generated by transducing cell lines of interest with lentivirus, containing pLKO.1 vector with non-targeting control shRNA or shRNA sequences targeting either *EBP* or *CHRM1* (purchased from Duke Functional Genomics Shared Resource, shRNA sequences listed in Appendix A) at an MOI of 1 or 3. CRISPR/Cas9-mediated gene knockout cells were generated by transducing BTIC#102 cells with *HRH1* and *CHRM3* CRISPR/Cas9 knockout lentivirus. Luciferase-expressing cells were generated by transducing C266 cells with CMV-Firefly luciferase lentivirus (Cellomics Technology, #PLV-10003-50, Halethorpe, MD, USA) at an MOI of 1. All transduction reactions were supplemented with 6 µg/mL polybrene (MilliporeSigma, #TR-1003, Darmstadt, Germany). Virus-containing media were removed after 24 h of transduction and replaced with the appropriate media. Transduced cells were selected with puromycin (MilliporeSigma, #P8833, Darmstadt, Germany) at 1 µg/mL for 5 days and subsequently maintained in 0.3 µg/mL puromycin.

### 2.2. Chemicals, Other Reagents and Plasmids

Chemicals and other molecular biology reagents that were used are listed in Appendix A. Briefly, stock solutions of all remyelinating agents were prepared in dimethyl sulfoxide (DMSO), with clemastine and benztropine prepared at 10 mM and other remyelinating agents at 20 mM, and stored at −20 °C. The 200 µM working solutions were then prepared by diluting the 10 mM stock solutions with human or mouse standard NSC media and stored at −20 °C. CW3388 stock solutions were prepared in dimethyl sulfoxide (DMSO) at 2 mM and stored at 4 °C. Lathosterol stock solutions were prepared in ethanol at 12.5 mM, water-soluble cholesterol stock solutions were prepared in PBS at 4 mM, and stock solutions of H1R inhibitors were prepared in DMSO at 10 mM.

The pcDNA3.1 V5-His A plasmid was a kind gift from Dr. Vidyalakshmi Chandramohan, and the pcDNA3.1+-C-(K)-DYK plasmid expressing Flag-EBP was purchased from GenScript (#OHu18817, Piscataway, NJ, USA). Primers for site-directed mutagenesis of the *EBP* gene were designed using the QuikChange Primer Design Program and listed in Appendix A. Three EBP mutant constructs (E80K, R147H, W196S) were constructed from pcDNA3.1+-C-(K)-DYK-*EBP* plasmids using the QuikChange Lightning Multi Site-Directed Mutagenesis Kit (Agilent Technologies, #210519, Santa Clara, CA, USA) following the manufacturer’s protocol. All plasmid sequences were verified by Sanger sequencing provided by Genewiz (South Plainfield, NJ, USA).

For CRISPR/Cas9-mediated gene knockout plasmids, single guide RNA (sgRNA) sequences targeting the exons of *HRH1* and *CHRM3* genes were designed using the CRISPOR web tool (http://crispor.tefor.net/, accessed on 13 September 2020) and are listed in Appendix A. *HRH1* and *CHRM3* CRISPR/Cas9 knockout plasmids were constructed from LentiCRISPRv2E plasmids by first phosphorylating and annealing paired sgRNAs with 10× T4 ligation buffer (New England Biolabs, #B0202S, Ipswich, MA, USA) and T4 polynucleotide kinase (New England Biolabs, #M0201S, Ipswich, MA, USA), and then ligating the annealed sgRNAs and pre-digested and purified LentiCRISPRv2E vector with the Quick Ligation Kit (New England Biolabs, #M2200S, Ipswich, MA, USA). Plasmids were verified to contain expected sgRNA sequences by colony PCR (using LKO.1 5′ and pLentiCRISPR-R1 primers) and/or Sanger sequencing (Genewiz, using T7 primers, South Plainfield, NJ, USA) using PCR primers listed in Appendix A. *HRH1* and *CHRM3* CRISPR/Cas9 knockout lentivirus was prepared by transfecting HEK293FT cells with Lipofectamine 3000 transfection reagent (Thermo Fisher Scientific, #L3000015, Waltham, MA, USA) and plasmids including pLP1, pLP2, pLP/VSVG, and two LentiCRISPRv2E plasmids containing different sgRNA sequences targeting the same gene (*HRH1* or *CHRM3*). Control lentivirus was prepared parallelly using LentiCRISPRv2E plasmids containing control sgRNA sequences.

### 2.3. Plasmid Electroporation

Cells were transfected with pcDNA3.1 V5-His A, pcDNA3.1+-C-(K)-DYK plasmids encoding either wild-type EBP or one of the three mutant EBPs (E80K, R147H, W196S), EGFP-hGal3, or PM-GFP plasmids using Neon Transfection System 10 μL Kit (Thermo Fisher Scientific, #MPK1025, Waltham, MA, USA) following the manufacturer’s protocol. In brief, cells were prepared and resuspended in buffer R at a concentration of 100,000 cells/10 µL buffer R. Then, 10 µL of cell suspension was added to each plasmid solution containing 0.5 µg or 1 µg of plasmid DNA. The cell-plasmid DNA mixtures were then subjected to electroporation using the Neon Transfection System and transferred into laminin-coated 24 well plates containing 500 µL of prewarmed standard NSC media in each well. Transfected cells were selected with G418 sulfate (ThermoFisher Scientific, #10131035, Waltham, MA, USA) at 250 µg/mL for 5 days and subsequently maintained in 100 µg/mL G418 sulfate.

### 2.4. BTIC Proliferation Assay, Cell Cycle Analysis and ELDA

Cell proliferation assays were performed by seeding the cell lines of interest in their respective media, with either vehicle or drugs of interest and without any antibiotics, at a density of 1500–3000 cells/well and a final volume of 200 µL/well in laminin-coated 96-well clear flat bottom plates. Cells were incubated overnight in a humidified incubator at 37 °C and 5% CO_2_, and then placed in the IncuCyte S3 Live-cell analysis system (Sartorius, Göttingen, Germany) and scanned every 4 h using the whole well, phase contrast acquisition mode (4× objective) for 7–14 days. Phase area confluence was obtained and calculated by the IncuCyte system, normalized to day 0 to generate relative phase object area (fold change), and then presented as mean ± S.E.M.

For cell cycle analyses, cells were re-suspended in 500 µL of ice-cold PBS and triturated to obtain a single cell suspension. Ethanol fixation was performed by first adding 3 mL of ice-cold 70% ethanol to the cell suspension in a drop-wise manner while vortexing, and then incubating the cell-ethanol mix at −20 °C for 1 h. The fixed cells were washed with ice-cold PBS three times, resuspended in 500 µL FxCycle™ PI/RNase Staining Solution (ThermoFisher Scientific, #F10797, Waltham, MA, USA), and incubated at room temperature for 30 min while protected from light. Samples were analyzed with BD Fortessa X-20 with the BD FACSDiVa software (BD Biosciences, San Jose, CA, USA). ELDAs with or without pretreatment were performed as previously described [43,44].

### 2.5. In Vivo Drug Treatment

Orthotopic intracranial tumor implantation and in vivo bioluminescent imaging were performed as previously described [43,44]. Briefly, C266 mouse glioma cells (~100,000 cells, expressing luciferase) were mixed 1:3 with methylcellulose and injected into the right caudate nucleus of female athymic nude mice (Jackson Labs, strain #:007850, Bar Harbor, ME, USA), and tumor treatment was initiated 4 days post-implantation. Mice were treated with vehicle control or clemastine at 30 mg/kg via intraperitoneal (i.p.) injection daily for five days per week (15% DMSO in PBS was used as the vehicle control). In vivo drug response was monitored weekly by bioluminescent imaging of mice and analyzed using the Living Image software (PerkinElmer, Waltham, MA, USA). Mouse body weight was documented at least every two days. All animal experiments were performed in accordance with protocols approved by the Duke University Institutional Animal Care and Use Committee (IACUC), Protocol #A133-19-06 (approved 27 June 2019).

### 2.6. Reverse Transcription and Real-Time Quantitative PCR (RT-qPCR)

BTIC cultures were plated at a density of 80,000–120,000 cells/well in laminin-coated 6 well plates in standard NSC media and incubated overnight to allow them to adhere. On the next day, cells were treated with either vehicle or drugs. Cells were passaged every 5–7 days into fresh media containing respective treatments, and then harvested on day 9–14 for total RNA extraction using the Quick-RNA Miniprep kit (Genesee Scientific, #11-328, El Cajon, CA, USA) following the manufacturer’s protocol. The concentration and the A260/A280 ratio of each RNA sample was measured with a Nanodrop Lite Spectrophotometer (Thermo Fisher Scientific, #ND-LITE, Waltham, MA, USA). RNA samples with concentration > 30 ng/µL and A260/280 ≥ 2.0 were then reverse-transcribed to cDNA using the EcoDry cDNA synthesis kit (Takara Bio, #639548, Shiga, Japan) following the manufacturer’s protocol. Real-time quantitative PCR was performed using gene-specific qPCR primers listed in Appendix A and the KAPA SYBR Fast qPCR Master Mix (2×) kit (Kapa Biosystems, #KK4602, Wilmington, MA, USA), and ran on the CFX96 Real-Time PCR Detection System (BIO-RAD, Hercules, CA, USA). The results were analyzed using the CFX Maestro Software (BIO-RAD, Hercules, CA, USA). The housekeeping genes *ACTB*, *B2M*, and *GAPDH* were used as internal controls for gene expression normalization.

### 2.7. Protein Extraction and Immunoblotting

Total cellular protein was extracted from cell pellets using 1% SDS lysis buffer. Briefly, frozen cell pellets were resuspended with 80–120 µL 95 °C preheated SDS lysis buffer (1% SDS, 50 mM NaF, 1 mM Na_3_VO_4_ in PBS, pH 7.4) and triturated until the suspensions were less gluey. The cell suspensions were then sonicated with Bioruptor Standard (Diagenode, #UCD-200, Denville, NJ, USA) for 5 min at high intensity to achieve complete cell lysis, and heated at 95 °C for 5 min. Cell lysates were centrifuged to remove residual cell debris, and supernatants, which were the protein extracts, were collected into new Eppendorf tubes. Protein concentrations were determined using the Pierce BCA Protein Assay kit (Thermo Fisher Scientific, #23225, Waltham, MA, USA) following the manufacturer’s protocol. The residual protein extracts were mixed with 4× Laemmli sample buffer (BIO-RAD, #1610747, Hercules, CA, USA) at a ratio of 1:3, heated at 95 °C for 5 min, cooled on ice for 5 min, and stored at −20 °C or used immediately for gel electrophoresis.

Immunoblotting was performed by loading 10–15 µg of protein samples or 3–5 µL of Precision Plus Protein Dual Color Standards (BIO-RAD, #1610374, Hercules, CA, USA) in NuPAGE 4–12% Bis-Tris protein gels (Thermo Fisher Scientific, #NP0321BOX, NP0322BOX, NP0335BOX, NP0336BOX, Waltham, MA, USA) and ran through the NuPAGE MOPS SDS Running Buffer (Thermo Fisher Scientific, #NP0001, Waltham, MA, USA) supplemented with 500 µL of NuPAGE Antioxidant (Thermo Fisher Scientific, #NP0005, Waltham, MA, USA). After gel electrophoresis, proteins were transferred from gels to nitrocellulose membranes (BIO-RAD, #1620115, Hercules, CA, USA) by semi-dry transfer methods using the Trans-Blot Turbo Transfer System (BIO-RAD, #1704150, Hercules, CA, USA). The membranes were then blocked with 5% non-fat dry milk (Genesee Scientific, #20-241, El Cajon, CA, USA) in TBST (0.1% Tween 20 in TBS) for 1 h at room temperature, and incubated overnight at 4 °C with primary antibodies diluted in antibody dilution buffer (5% BSA, 0.02% NaN_3_ in TBST) according to the manufacturer’s suggested dilutions (detailed information for antibodies and their dilutions were listed in Appendix A). On the next day, the membranes were incubated with appropriate horseradish peroxidase-conjugated secondary antibodies (Cell Signaling Technology, anti-rabbit IgG, #7074S; anti-mouse IgG, #7076S, Danvers, MA, USA) diluted in TBST for 1 h at room temperature. Then, the chemiluminescent signals were enhanced by incubating the membranes with SuperSignal West Pico PLUS Chemiluminescent Substrate (Thermo Fisher Scientific, #34580, Waltham, MA, USA) following the manufacturer’s protocol. The membranes were imaged with ChemiDoc MP System (BIO-RAD, #1708280, Hercules, CA, USA) and analyzed with Image Lab Software (BIO-RAD, Hercules, CA, USA). To re-probe the membranes, membranes were stripped using a Restore Western Blot Stripping Buffer (Thermo Fisher Scientific, #21059, Waltham, MA, USA) and incubated with other primary antibodies following the manufacturer’s protocol.

### 2.8. Immunofluorescence Staining

Cells were plated in laminin-coated 6 well plates and treated with either vehicle or clemastine (4 µM), as previously described. After 13–26 days of treatment (noted in the figure legends), the cells were plated on laminin-coated Nunc Lab-Tek II 2-well chamber slides (Thermo Fisher Scientific, #154461, Waltham, MA, USA) or 3-well removeable chamber slides (Ibidi, #80381, Gräfelfing, Germany) and grown until they reached 80–90% confluence. Cells were washed with PBS, fixed in 10% Neutral buffered formalin (VWR, #89370-094, Radnor, PA, USA) or 4% formaldehyde (1:4 diluted from 16% formaldehyde with PBS) at room temperature for 15 min, and permeabilized with 0.1% saponin (MilliporeSigma, #84510, Darmstadt, Germany, only in samples stained for GALC) or 0.3% Triton X-100 (MilliporeSigma, #93443, Darmstadt, Germany) for 10 or 15 min, respectively. Slides with fixed and permeabilized cells were blocked with blocking buffer (1% BSA, 0.1% saponin or 0.2% Triton X-100, 10% goat serum in PBS) at room temperature for 1 h, and then incubated with primary antibodies (detailed information of antibodies are listed in Appendix A) diluted in antibody dilution buffer (1% BSA, 0.1% saponin or 0.2% Triton X-100 in PBS) at 4 °C overnight. The next day, slides were washed three times with 0.1% saponin or 0.2% Triton X-100 in PBS, and incubated with appropriate secondary antibodies (anti-rabbit Alexa 594, #A-11037; anti-rabbit Alex 647, #A-21245; anti-mouse Alexa 488, #A-11029, Thermo Fisher Scientific, Waltham, MA, USA) 1:500 diluted in antibody dilution buffer at room temperature for 1 h. Cells were stained with 1 µg/mL DAPI (MilliporeSigma, #D9542, Darmstadt, Germany) before the slides were mounted with ProLong Gold Antifade Mountant (Thermo Fisher Scientific, #P36394, Waltham, MA, USA) or SlowFade Diamond Antifade Mountant (Thermo Fisher Scientific, #S36972, Waltham, MA, USA) following the manufacturer’s protocol. The slides were imaged with Zeiss Axio Imager Z2 Upright Microscope and analyzed using the Zeiss Zen 3.5 (blue edition) software (Carl Zeiss Microscopy GmbH, Oberkochen, Germany).

Images from PDGFRA and GALC immunofluorescence staining of the vehicle or clemastine-treated BTIC#102 cells were analyzed using the CellProfiler software (version 4.2.1, Broad Institute, Cambridge, MA, USA) [47]. For quantification of mean fluorescence intensity of PDGFRA and GALC, “Cell” objects were first identified using the DAPI signal for nuclear regions and the PDGFRA/GALC signal to outline cell borders. The mean fluorescence intensity of PDGFRA and GALC per “Cell” object was measured, and the mean of mean fluorescence intensity per “Cell” object per field of view was then calculated. For quantification of percent positivity of PDGFRA and GALC, PDGFRA^+^ and GALC^+^ “Cell” objects were first identified using the mean fluorescence intensity per “Cell” object as thresholds, which were set using isotype-stained images as negative controls. The percentage of PDGFRA and GALC positive cells were then calculated as $\frac{number\;of\;positive\;Cell\;object}{total\;number\;of\;Cell\;Object}\;$ per field of view. To quantify the frequency distribution of PDGFRA^+^ and/or GALC^+^ cells, GALC positivity was determined in every PDGFRA^−^ and PDGFRA^+^ cell from all fields of view regardless of treatment status, and the contingency table was generated based on the calculated frequency distribution.

### 2.9. RNA-Seq Data and Pathway Analysis

Next generation sequencing for mRNA-seq, including library construction and sequencing, was provided by Novogene Corporation Inc. (Sacramento, CA, USA). NovaSeq 6000 was used for PE150 sequencing. mRNA-seq data were analyzed using the Galaxy web platform via the public server at usegalaxy.org [48]. The workflow of RNA-seq analysis was adapted from a previously described procedure [49]. Briefly, raw data were trimmed by Trim Galore, aligned by Hisat2, re-assembled by StringTie, and differentially expressed genes (DEGs) were analyzed by Deseq2. All parameters were set at default. For pathway enrichment analysis, differentially expressed genes (DEGs, sorted by adjusted *p*-value < 0.05) were imported into ShinyGO [50] (http://bioinformatics.sdstate.edu/go/, accessed on 14 October 2021). For pathway analysis, KEGG and Panther pathways were used, and genes with adjusted *p*-values > 0 were input as the background gene list. For GSEA, all genes or genes with FDR ≤ 0.25 from human or mouse mRNA-seq data, respectively, were ranked by fold change and subjected to GSEA preranked analyses following the established protocol [51] using the GSEA software (version 4.1.0, Broad Institute, Cambridge, MA, USA). For pathway impact (two-evidence) analysis, all genes with adjusted *p*-values ≤ 0.05 were selected and subjected to analysis using ROntoTools [52], and pathways were ranked by combined FDR *p*-value (pComb.fdr).

### 2.10. Statistical Analysis

All in vitro proliferation assays, cell cycle analyses, ELDA, quantitative RT-PCR, immunoblotting, immunofluorescence staining, and in vivo clemastine treatment experiments were repeated in at least two independent experiments. The number of independent samples was noted in the figure legends. Data were presented as mean ± standard error of the mean (S.E.M.), except for gene expression data, which were presented as geometric mean ± geometric standard deviation (S.D.), and ELDA data, which were shown as trend lines ± two-sided 95% confidence intervals. Mean values between two groups were compared by student *t*-tests (with Welch’s correction when variances were deemed significantly different by F tests) or non-parametric Mann–Whitney tests. Mean values between 3 or more groups were compared by one-way or two-way ANOVA followed by Dunnett’s or Tukey’s multiple comparisons tests, respectively. Mean values between two or multiple groups over time were compared to the control group by two-way repeated measures ANOVA followed by Sidak’s or Dunnett’s multiple comparisons tests, respectively. Mean values of multiple groups were compared between all possible group pairings by two-way repeated measures ANOVA, followed by Tukey’s multiple comparisons tests. Geometric mean values of gene expression fold change were log_2_-transformed and analyzed by one-sample *t*-tests compared to zero. Survival analyses were performed using Log-rank (Mantel-Cox) tests. Contingency tables were analyzed using one-sided chi-square tests. All tests were two-sided, if not otherwise specified, and deemed statistically significant when *p*-values < 0.05. Normality was tested before conducting any parametric test using Shapiro–Wilk normality tests. For proliferation assays, significance was calculated using the data from the last timepoints unless stated otherwise. All statistical analyses were performed using GraphPad Prism software (version 9.3.1, San Diego, CA, USA) except for ELDA (analyzed by ELDA online software: https://bioinf.wehi.edu.au/software/elda, accessed on 17 April 2021), Gliovis gene expression data (analyzed by Gliovis portal: http://gliovis.bioinfo.cnio.es/, accessed on 18 October 2021), RNA-seq, GSEA, and pathway analyses.

## 3. Results

### 3.1. Clemastine Suppresses the Propagation of Patient-Derived BTIC Cultures

Several drugs with OPC-differentiating activity have been shown to effectively stimulate normal OPC differentiation and subsequent remyelination in multiple in vivo disease models [40,41,42]. These findings prompted us to test the effects of these drugs on GBM cells bearing OPC features. The PN subtype of GBM, typified by OPC-like transcriptomes, is characterized by IDH1 mutations or *PDGFRA* amplification, while *EGFR* amplification associates with the CL subtype [9]. Therefore, we first identified GBM lines with predominantly *EGFR* amplification (*EGFR^+^*, representing CL GBMs), both *PDGFRA* and *EGFR* amplifications (in agreement with the heterogeneous nature of GBMs that are observed), as well as lines that bear predominantly *PDGFRA* amplification (*PDGFRA*^+^, representing PN GBMs) (Appendix A). We then cultured two *PDGFRA*^+^ lines (BTIC#102 and BTIC#148, with ~25× and ~3× *PDGFRA* amplification, respectively) in serum-free neural stem cell medium to maintain their stemness properties as BTIC cultures, and determined the effects of a small panel of OPC-differentiating drugs on BTIC propagation. We found that all of these drugs suppressed the propagation of BTICs in a dose-dependent manner (Appendix A). Among them, tamoxifen, benztropine, and clemastine were previously found to promote OPC differentiation by affecting the cholesterol biosynthesis pathway [53]. Of note, clemastine is a safe, over-the-counter (OTC) allergy relief medicine that can cross the blood–brain barrier (BBB) and induce remyelination [40,41,54]. It has shown promising therapeutic benefits via differentiation induction in demyelinating models [40,41,42] and, most remarkably, in clinical trials for MS patients [54,55]. These findings prompted us further to characterize the inhibitory effects of clemastine. We found that the effects of clemastine on BTICs were accompanied by cell morphology alterations (Figure 1A,B and Appendix A) and suppressed cell proliferation (Figure 1C and Appendix A). Notably, following 10-day clemastine pre-treatments, the antiproliferative effects of clemastine persisted even after clemastine was removed from the media (Figure 1D and Appendix A). Collectively, these results suggest that clemastine induced lasting alterations in *PDGFRA*^+^ BTICs and suppressed the propagation of these tumor cells.

While we had focused on PN-like BTICs that are believed to serve broadly as progenitors of GBMs [16] and transition between one subtype to another (e.g., PN to MES GBMs) [14,15], the intratumoral heterogeneity of GBMs consisting of various subtypes [7,8,9,10,11,12] prompted us to examine the effects of clemastine in primary BTIC cultures resembling other subtypes. We used two primary *EGFR^+^* BTIC cultures (representing the CL subtype: BTIC#127 and BTIC#095) (Appendix A) and two primary BTIC cultures that were shown to bear MES transcriptomic signatures (GSC-1123 and JK-046 [45,46]) for this purpose. We found that BTIC cultures resembling the CL and MES subtypes were also susceptible to clemastine treatment, as indicated by reduced cell proliferation (Figure 1E), suggesting that the anti-proliferative effects of this drug impact BTICs broadly and are independent of tumor subtypes. Importantly, unlike the tumor cells tested, the proliferation of normal human astrocytes was not affected by the treatment (Figure 1F). Collectively, these results implicate clemastine as a BTIC-suppressive drug and provide a basis for repurposing clemastine, and potentially other remyelinating agents, for targeting GBMs.

### 3.2. Clemastine Attenuates the Stemness and Progenitor Cell Features of PDGFRA^+^ BTICs

Clemastine has been shown to promote the differentiation of OPCs to mature oligodendrocytes in treating MS [53,56]. This prompted us to investigate whether the effects of clemastine on BTICs involved such a differentiation induction mechanism. We used *PDGFRA*^+^ BTIC cultures as the prototypic models with the rationale that the OPC-like status of these tumor cells can facilitate assessing their progenitor versus differentiated states. We treated the *PDGFRA^+^* BTIC cultures with clemastine and evaluated the expression of key markers indicative of neural stem cell (NSC) and OPC properties. In agreement, clemastine treatment resulted in the decreased expression of marker genes for NSCs (*NES* [57], *SOX2* [58]), OPCs (*OLIG2* [59], *PDGFRA* [60], *CSPG4* [61]), and negative regulators of OPC differentiation (e.g., *NOTCH1*, *CSPG5* [62]) in the two *PDGFRA*^+^ BTIC cultures (Figure 2A and Appendix A). Intriguingly, *MOG*, an oligodendrocyte maturation marker gene [63], displayed upregulated expression in clemastine-treated cells (Figure 2A). The reduced expression of these molecular markers of NSCs and OPCs was confirmed at their protein levels by immunoblots (Figure 2B and Appendix A) and immunofluorescent staining (Figure 2C and Appendix A). To further investigate cell state changes following clemastine treatment, we utilized BTIC#102, the line with a high level of *PDGFRA* amplification and PDGFRA expression (i.e., more OPC-like), to facilitate studying the differentiating effects. The diminished progenitor states and potential differentiation of tumor cells were corroborated by the induction of GALC protein expression, another oligodendrocyte maturation marker [63], in parallel with the reduced level of PDGFRA protein (Figure 2D,E). The increased presence of GALC^+^ and reduced presence of *PDGFRA*^+^ cells (Figure 2F), together with the mutually exclusive staining pattern of these proteins (Figure 2G), suggest these tumor cells indeed became more differentiated upon clemastine treatment.

The altered expression of marker genes associated with progenitor or differentiated cells prompted us to examine the changes in the global gene expression (mRNA-seq) of the two *PDGFRA*^+^ BTIC cultures in response to clemastine. Consistent with our observations mentioned above, transcriptomic profiling from the mRNA-seq data revealed an overall downregulated expression of most dominant marker genes for OPCs [64] in response to clemastine treatment in both BTIC cultures (Figure 2H and Appendix A). mRNA-seq analyses also revealed upregulated expression of a fraction of marker genes for newly formed oligodendrocyte (NFO) and myelinating oligodendrocytes (MO) [64] in response to clemastine treatment (Figure 2I and Appendix A). Gene Set Enrichment Analysis (GSEA) confirmed the clemastine-induced downregulation of genes associated with OPCs (Appendix A) [65], and downregulation of genes in the protein translation machinery (Appendix A), which serves a critical role in cancer cell stemness/plasticity and as a promising therapeutic target [66,67].

To assess the functional consequence of the altered gene expression profiles described above, we determined the effects of clemastine on the stem-like properties of BTICs, as measured by their self-renewal capability [68,69], via the Extreme Limiting Dilution Assay (ELDA) [70]. We found that clemastine treatment diminished the self-renewal capability of BTICs (Figure 2J). More notably, reminiscent of the lasting effects of clemastine on BTIC propagation, BTICs that were exposed to clemastine and then maintained in clemastine-free media also displayed a diminished self-renewal capacity, suggesting a persistent effect of clemastine on BTIC stemness (Figure 2K). Together with the altered gene expression profiles, these results suggest that clemastine attenuates the stemness/progenitor properties of BTICs by promoting differentiation.

### 3.3. EBP, A Pharmacological Target of Clemastine, Is Essential for BTIC Propagation

Pharmacologically, clemastine is known to act as an antagonist for histamine H1 receptor (H1R, encoded by *HRH1*) and muscarinic receptors (M1R-M5R, encoded by *CHRM1*-*CHRM5,* respectively) [71], and to target EBP, an enzyme in the cholesterol biosynthesis pathway. The well-established antagonist activity of clemastine against H1R and the inhibitory effects of M1R and M3R signaling on oligodendrocyte differentiation [72,73] prompted us to determine the expression profiles of these three receptor genes (*HRH1*, *CHRM1,* and *CHRM3*) in gliomas. We found that *HRH1* and *CHRM3* were expressed at higher levels in GBM (grade IV) in comparison to lower-grade gliomas (grade II and III), while *CHRM1* displayed the opposite trend (Appendix A). This result led us to experimentally assess the roles of H1R and M3R in BTICs. First, two second-generation H1R-antihistamines, fexofenadine (Allegra) and cetirizine (Zyrtec) [74], were found to have no suppressive effects on BTICs, suggesting these cells were not affected by the H1R signaling blockade (Appendix A). Furthermore, CRISPR/Cas9-mediated individual gene knockout of *HRH1* or *CHRM3* (Appendix A) did not affect the propagation of BTICs or the susceptibility of BTICs to clemastine (Appendix A). As the tumor-supporting roles of each gene may be affected by the variable genetic makeups specific for each tumor model, these results did not conclusively rule out the involvement of these receptors in mediating the effects of clemastine. Nevertheless, these findings led us to investigate other potential targets of clemastine. Interestingly, the pharmacological inhibition of EBP has been shown to effectively induce the differentiation of OPCs to become oligodendrocytes [53,56], which prompted us to examine the roles of EBP in GBM (Appendix A).

First, analysis of mRNA-seq data from The Cancer Genome Atlas (TCGA) [75] via the Gliovis data portal [76] revealed that, distinctively, from genes encoding other enzymes in the cholesterol biosynthesis pathway, EBP displayed an upregulated expression in GBMs in comparison to non-tumor tissues, as well as to lower grade gliomas (grade II and III) (Appendix A). Notably, two additional properties distinguish EBP from other enzymes in the cholesterol biosynthesis pathway: EBP mutations have been linked to Conradi-Hünermann-Happle (CHH) syndrome [77], and it is known to be a protein target of numerous structurally diverse pharmacological agents [78,79,80]. In fact, EBP is the common pharmacological target shared by seven out of the eight OPC-differentiating drugs included in our initial test [53,56,78,79] (Appendix A). These unique features, together with the therapeutic potential of targeting EBP, led us further to assess the pathological functions of this protein in GBM.

We used the patient-derived BTIC cultures to determine the roles of EBP in GBM. The knockdown of EBP in BTICs resulted in significant suppression of BTIC propagation, which was accompanied by altered cell morphology (Figure 3A,B and Appendix A) and reduced expression levels of featured OPC marker genes, including *PDGFRA*, *CSPG4,* and *OLIG2* (Appendix A). Conversely, echoing the finding from the loss-of-function assays, an overexpression of exogenous EBP promoted BTIC propagation (Figure 3C and Appendix A). Several pathogenic, missense mutations of EBP have been identified in CHH patients with various degrees of reduction in enzymatic activities [77,81,82]. Therefore, we assessed if enzymatic activity is required for the promoting effect of EBP by overexpressing three of the EBP mutants, EBP-E80K, EBP-R147H, EBP-W196S, or the wild-type EBP in BTIC#102 cells (Appendix A). Of note, levels of these mutant EBP proteins were lower than the wild-type EBP protein (Appendix A), likely due to their various stabilities. Nevertheless, among the three mutants, which had comparable expression levels, EBP-E80K, the mutant known to have minimal enzymatic activity retained, had lower potency in promoting the growth of the GBM culture compared to EBP-R147H, a mutant that possesses partial enzymatic activity [77,83] (Figure 3D). Taken together, these results suggest that EBP is essential for BTIC propagation and that the GBM-promoting effects of EBP likely require its enzymatic activity.

Several additional lines of evidence corroborate the findings mentioned above and support that suppressing EBP partially mediates the effects of clemastine on BTICs. First, the addition of exogenous cholesterol to the culture media partially rescued the suppressive effects of clemastine on BTIC propagation (Figure 3E and Appendix A). Furthermore, the presence of the immediate metabolic product of EBP, lathosterol, also partially countered the suppressive effects of clemastine on BTIC propagation. Of note, the addition of exogenous lathosterol alone at a higher dose moderately suppressed tumor cell propagation (Figure 3F and Appendix A), suggesting that an optimal dose of this metabolite and maintaining the homeostasis of the cholesterol pathway are essential for the optimal growth of tumor cells. Finally, we found that a stable overexpression of EBP in primary BTICs conferred partial protection from clemastine (Figure 3G). Collectively, the above results suggest that BTIC propagation depends on EBP, a pharmacological target of clemastine, and the homeostasis of the cholesterol pathway, and provide a potential mechanism underlying the suppressive effects of clemastine.

### 3.4. Clemastine Suppresses Tumorigenicity in A Mouse Glioma Model Representative of PN GBMs

We postulated that a GBM model with a better-defined genetic makeup could provide advantages to further characterize the tumor suppressive effects of clemastine. For this purpose, we generated mouse NSC lines from a conditional *Trp53* knockout mouse model previously described [84,85]. NSCs derived from these models underwent adenoviral Cre-recombinase mediated *Trp53* deletion followed by the overexpression of exogenous *Pdgfb*, an oncogene that has been shown to induce gliomas that resemble PN GBMs [86,87,88,89,90]. The genetically modified NSC lines were then orthotopically implanted into mice to generate PDGFB-driven, aggressive glioma tumors with high penetrance and short latency (Appendix A). Gene expression profiling (mRNA-seq) of the tumor tissues and subtyping analysis utilizing the previously defined subtype-specific gene signatures [90] confirmed their predominant PN features, and the detectable expression of genes associated with other subtypes recapitulated the heterogeneous nature of GBMs (Appendix A). A glioma cell line derived from these tumors (named C266) was used to investigate its dependency on the EBP protein and susceptibility to clemastine, benztropine, and a small-molecule drug (CW3388) known to inhibit mouse EBP [56]. Results from these experiments using the mouse glioma cell line corroborate findings from human BTICs, as elaborated in the following:

First, the knockdown of *Ebp* resulted in attenuated growth (Figure 4A and Appendix A) and reduced self-renewal capacity (Figure 4B) of the mouse glioma cell line. Second, treating C266 cells with CW3388 resulted in decreased cell proliferation (Figure 4C). Similarly, the mouse glioma line also displayed suppressed propagation in response to clemastine and benztropine treatment, recapitulating the findings from human BTICs (Figure 4D and Appendix A). Third, echoing the effect of EBP’s abundance on human BTIC’s susceptibility to clemastine, *Ebp* knockdown rendered mouse glioma cells more sensitive to clemastine and CW3388 (Figure 4E and Appendix A), suggesting that clemastine acted in the same manner as CW3388. Taking advantage of the aggressive nature of this tumor cell line in orthotopic mouse models, we tested the suppressive effects of clemastine in vivo. We found that the clemastine treatment led to a delayed progression of tumors in vivo, as evidenced by the slower tumor progression (Figure 4F), and a small subset of mice (20%) that had extended survival with measurable tumor signal two months post-implantation (at which point, mice were terminated) (Figure 4G). The reasons for this delayed yet extended benefit remained unclear, and we speculate that the treatment likely altered the progenitor properties of tumor cells instead of acutely killing them in vivo. Finally, we noted that mice subjected to this treatment regimen displayed no different body weight compared to the control mice, suggesting its minimal toxicity (Appendix A).

### 3.5. Clemastine Treatment Broadly Alters Multiple Signaling Pathways in Glioma Cells

While clemastine has multiple pharmacological targets and its effects on cell signaling pathways are expected to be broad and dependent on the genetic composition of GBM cells, we postulated that identifying pathways that were perturbed by clemastine and/or CW3388 in the mouse glioma cell line could provide additional insights into the molecular mechanisms of clemastine. Therefore, we performed mRNA-seq to identify differentially expressed genes in clemastine or CW3388-treated C266 cells (Appendix A). Several findings were noted. First, among the 10,292 genes identified, 18.3% (1888 genes) of them were affected by clemastine. In comparison, among those genes affected by CW3388 (2759 genes), the number increased to 45.7% (1261 genes), suggesting genes affected by clemastine were enriched in the CW3388-responsive gene set (*p* < 0.0001). Additionally, among genes affected by both agents, an overwhelming majority of them (1210 out of 1261; 96.0%) displayed the same directional changes (i.e., up- or down-regulation) in response to both agents, further supporting that clemastine and CW3388 act on overlapping genes/pathways in the tumor cells. Second, GSEA of differentially expressed genes presented a positive enrichment of genes corresponding to oligodendrocytes [91] in response to either clemastine or CW3388, in agreement with their expected effects on a tumor cell’s progenitor identity (Appendix A). Third, in clemastine-treated cells, KEGG pathway analysis identified pathways in cancers, metabolism, Wnt, and PI3K-AKT signaling pathways (Appendix A). Notably, similar analyses comparing the CW3388-treated versus the vehicle-treated cells identified an overlapping set of KEGG pathways (Appendix A). Panther pathway analysis pinpointed smaller numbers of altered pathways using the same criteria (FDR ≦ 0.05), and revealed that a majority of the CW3388-affected pathways were also identified as being perturbed by clemastine (Appendix A). These results suggest that the impacts of clemastine overlap with and exceed those of CW3388, in agreement with its expected pharmacological activity targeting multiple proteins, including EBP.

We employed signaling pathway impact analysis, which incorporates an over-representation analysis and functional class scoring and was shown to have superior specificity and sensitivity in identifying altered pathways [52], to analyze differentially expressed genes of patient-derived BTIC cultures in response to clemastine (Appendix A). This analysis identified numerous KEGG pathways that were affected in the BTIC cultures (196 and 154 were identified in BTIC#102 and BTIC#148, respectively). Although the pathways varied between these models, likely due to the various genetic background and different basal gene expressions in matched control cells, 142 common pathways were identified in both BTIC cultures, including cell adhesion, cholesterol metabolism, and the PI3K-AKT signaling pathway (Appendix A). Collectively, these pathway analyses suggest the pleiotropic consequences of clemastine treatment at the molecular level, and also highlight the substantial alterations in cellular processes and oncogenic signaling pathways in tumor cells, including cell metabolism, the Wnt signaling pathway, and the PI3K-AKT pathway. While such assays could not distinguish pathway alterations that contributed to the tumor suppression from those that resulted from cell adaptation/population evolution, these results resonate with our findings in the mouse glioma cell line, and yield additional evidence to support the tumor suppressive effects of clemastine.

## 4. Discussion

BTICs are key drivers of GBM’s resistance to treatments, progression, and recurrence. In this study, we showed that clemastine, an over-the-counter drug used for alleviating allergies, suppresses the propagation of patient-derived BTICs bearing *PDGFRA* amplification (as a surrogate representing the PN GBMs), as well as those resembling CL or MES subtypes. We further used the PN/OPC-like BTICs to demonstrate that this drug attenuates the stemness/progenitor features of tumor cells. Finally, we showed that pharmacological targets of clemastine, as highlighted by EBP, are essential for the proliferation of these tumor cells.

The suppressive effects of clemastine on the BTICs described in the current study corroborate the findings from previous research on clemastine. First, it was shown that clemastine could compromise the lysosomal membrane integrity in glioma cells to suppress glioma cell tumorigenicity [92]. This study aligns with our findings in supporting that this drug can act on multiple pathways and cellular processes to exert tumor suppressive effects, and provide a foundation for investigating the detailed mechanisms in further studies. Second, the induction of normal OPC differentiation into myelinating oligodendrocytes by clemastine has yielded promising therapeutic benefits in clinical trials in MS patients, validating the in vivo therapeutic efficacy of clemastine and its permeability across the blood–brain barrier [54,55].

We note that, unlike in normal cells, genetic and epigenetic alterations in tumor cells most likely preclude the possibility of accomplishing terminal differentiation in the latter. Nevertheless, the suppressive effects of clemastine on GBM stemness could presumptively subvert the resistance mechanisms (e.g., heightened DNA repair machinery [93] and metabolic plasticity [94]), address the heterogeneity issue attributed to BTIC multipotency, and provide therapeutic benefits. Specifically, we propose to fully exploit the therapeutic potential of clemastine by combinatorial strategies with therapeutics such as (i) standard chemo- and radiation therapy; and (ii) drugs that target other subtypes of GBMs (e.g., anti-mesenchymal drug PTC596 [95], anti-classical drugs gefitinib [96] and nimotuzumab [97]). These strategies can potentially mitigate the emergence or selection of the evolved populations, such as those driven by EGFR oncogenic signaling [15].

The fact that an OTC medication, such as clemastine, demonstrates effectiveness in differentiating and attenuating stemness in BTICs offers an intriguing opportunity for GBM treatment development. In comparison with traditional chemo- and radiation therapy, the differentiation-based mechanism of clemastine is less dependent on proliferation status, and clemastine boasts the established BBB permeability and an incomparable safety profile with only minor adverse effects such as sedation and xerostomia (oral dryness). In support, many other OTC medications have gained sufficient experimental evidence and entered clinical trials for the investigation of GBM treatment (Appendix A [98,99,100,101,102,103]), and clemastine represents a promising therapeutic agent worthy of further investigation and awaiting to join the list.

The loss-of-function of EBP, a protein that is pharmacologically inhibited by clemastine [53,56], recapitulated the suppressive effects of clemastine, as indicated by both attenuated BTIC stemness and the accompanied differential gene expression. Taken together with the dependency of BTICs on EBP, our study implicates sterol metabolism in GBM stemness maintenance. Prior studies support that inhibitors for certain enzymes in the cholesterol biosynthesis pathway, including EBP, can promote remyelination, linking this pathway and the involved intermediate metabolites to promoting normal OPC differentiation [53,56]. In addition, lipid biosynthesis and the cholesterol pathway have been demonstrated to be promising therapeutic targets for treating GBMs [104,105,106,107,108]. For instance, intermediate metabolites in the cholesterol pathway were found to exert the tumor suppressive effects via activating endogenous liver X receptors (LXRs) in GBM [104,107] and another type of brain tumor, diffuse intrinsic pontine glioma (DIPG) [109]. Thus, our results are consistent with the previous literature that highlights the tumor cell’s dependency on the homeostasis of the cholesterol pathway.

Three outstanding issues and limitations remain to be further investigated and addressed. First, although EBP inhibition likely contributed to the suppressive effects of clemastine, we cannot rule out the potential promiscuity of clemastine and that additional pathways (e.g., histamine/muscarinic receptor signaling) might participate in mediating clemastine’s effects, as suggested by the remaining sensitivity of sh*Ebp* mouse glioma cells to clemastine (Figure 4E and Appendix A). The underlying mechanism of clemastine-mediated EBP inhibition also remains to be studied. This issue is complicated by EBP’s unique multi-drug binding capacity and the possibility that EBP lowers the intracellular concentration of drugs that inhibit its activity [83,110,111], which provide an alternative explanation for the findings in which the knockdown of EBP conferred the tumor cell’s sensitivity (and vice versa) to its inhibitors (such as clemastine and CW3388). Second, the essentiality of EBP in BTICs needs to be assessed and validated in vivo by performing limiting dilution experiments using the isogenic *Ebp*-knockdown mouse glioma cell line in our mouse models. Third, illustrating the molecular mechanisms of the tumor suppressive effects of EBP inhibition will be critical for assessing the potential of this enzyme as a therapeutic target. It is possible that EBP inhibition alters the intermediate metabolite composition in tumor cells (cell-autonomous) and/or in the tumor microenvironments (e.g., non-tumor cells). Alternatively, the upregulated expression of EBP in GBMs, a trend that is opposite to several other enzymes in the cholesterol biosynthesis pathway, as well as the complicated pharmacological property of this protein [78,79], raise the possibility that its inhibition affects cellular processes other than simply disrupting the biosynthesis pathway of cholesterol.

## 5. Conclusions

In summary, findings from this study support the feasibility of inducing differentiation for targeting the stemness of gliomas bearing OPC features, which include not only PN-like GBM, but also oligodendroglioma and DIPG [112,113]. We provide evidence to support exploiting clemastine, or other non-oncology, low-toxicity drugs with similar activity in targeting EBP and/or inducing differentiation [78,79,114], for targeting GBM stemness. In addition, the study also nominates EBP as a therapeutic target in GBM. During the preparation of this manuscript, the FDA has granted orphan drug designation (ODD) for DSP-0390, a newly developed inhibitor of EBP (https://www.targetedonc.com/view/fda-grants-orphan-drug-designation-to-dsp-0390-in-brain-cancer, accessed on 19 July 2022), for which a phase I clinical trial for treating recurrent high-grade gliomas has been initiated (NCT05023551) [115]. The results presented in this study provide evidence to support such actions. We propose that further studies assessing the underlying pathogenic mechanism of EBP and the therapeutic efficacy of EBP inhibitors on various subtypes of GBM cells in combination with other treatment approaches are warranted.

## Figures and Tables

**Figure 1 cancers-15-04619-f001:**
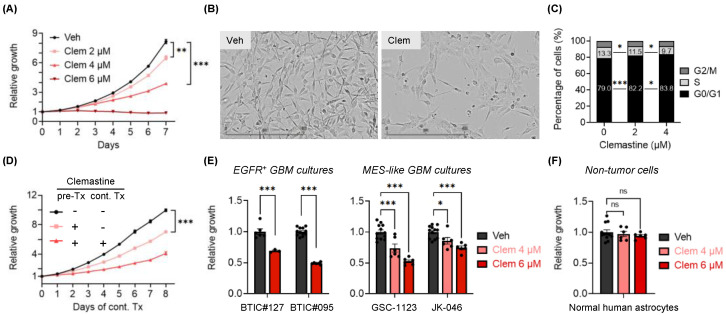
Clemastine suppresses the propagation of patient-derived BTIC cultures. (**A**) Proliferation of BTIC#102 cells treated with clemastine (Clem) at indicated doses. *n* = 3 per condition except vehicle, *n* = 9. (**B**) Representative images (4×, scale bar: 400 µm) of BTIC#102 cells treated with vehicle (Veh) or clemastine (Clem; at 4 µM) for 20 days in laminin-coated plates. (**C**) Quantification of cell cycle analysis of BTIC#102 cells treated with clemastine at indicated doses for 12 days. *n* = 2 per condition. (**D**) Proliferation of BTIC#102 cells with or without 10-day clemastine (4 µM) pre-treatments (pre-Tx) and/or subsequent clemastine (4 µM) treatments (cont. Tx). Cell proliferation was monitored during subsequent clemastine treatments. *n* = 9 per condition except for the pre-Tx^+^cont.Tx^+^ group, *n* = 6. “-“: no clemastine in the media; “+”: with clemastine in the media. (**E**,**F**) Quantification of relative cell proliferation of BTICs bearing *EGFR* amplification (**E**, left panel), or features of mesenchymal (MES) subtypes (**E**, right panel), or (**F**) normal human astrocytes treated with clemastine (Clem) at indicated doses. The *y*-axis represents normalized phase area confluence at day 4 (normalized to day 0 and then normalized to respective vehicle controls). *n* = 12 for all vehicle groups except BTIC#127, *n* = 6; *n* = 6 for all drug-treated groups except BTIC#127, *n* =3. (**A**, **D**–**F**) Data are represented as mean ± S.E.M. Significance was calculated using two-way repeated measures ANOVA followed by (**A**) Dunnett’s or (**D**) Tukey’s multiple comparisons tests or two-way ANOVA followed by (**C**) Tukey’s or (**E**,**F**) Sidak’s multiple comparisons tests, and was represented as * *p* < 0.05, ** *p* < 0.01, *** *p* < 0.001, ns: not significant.

**Figure 2 cancers-15-04619-f002:**
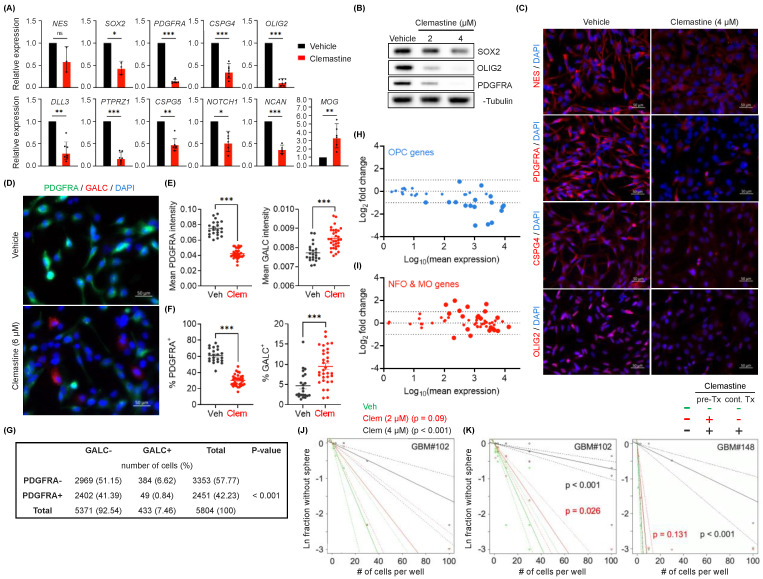
Clemastine attenuates the stemness and progenitor cell features of *PDGFRA*^+^ BTICs. (**A**) mRNA expression levels of genes associated with NSCs, OPCs, oligodendrocytes, and suppressors of OPC differentiation in BTIC#102 cells treated with vehicle or clemastine (4 µM) for 9–15 days as assessed by quantitative RT-PCR. *n* = 7 per gene except *MOG*, *CSPG5*, and *NOTCH1*, *n* = 6, *NCAN*, *n* = 5, and *NES*, *SOX2*, *n* = 3. (**B**,**C**) Protein levels of NSC and OPC markers in BTIC#102 cells treated with clemastine at indicated doses for 13–14 days were assessed by (**B**) immunoblot assays (representative results from three independent experiments) or (**C**) immunofluorescence (IF) staining (representative images from two independent experiments; 20×, scale bar: 50 µm). (**D**–**G**) IF staining of PDGFRA and GALC in BTIC#102 cells treated with vehicle or clemastine (Clem; at 6 µM) for 26 days. (**D**) Representative IF images (20×, scale bar: 50 µm), (**E**) quantification of mean PDGFRA and GALC fluorescence intensities, and (**F**) percentages of PDGFRA^+^ and GALC^+^ cells of the vehicle and clemastine-treated groups. (**E**,**F**) Each data point was calculated from a different field of view. *n* = 22 fields of view for all vehicle groups, and *n* = 32 for all clemastine-treated groups. (**G**) The contingency table of PDGFRA^−/+^ and/or GALC^−/+^ cell counts to assess the mutual exclusivity of PDGFRA and GALC positivity. A total of 5804 cells were identified from 54 fields of view. (**H**,**I**) MA plots summarizing differential mRNA expression of top 40 (**H**) OPC-specific genes or (**I**) oligodendrocytes-specific genes between clemastine versus vehicle-treated (15-day treatment) BTIC#102 cells. Larger symbols indicate genes with adjusted *p*-values < 0.05. (**J**) Extreme limiting dilution analysis (ELDA) of BTIC#102 cells treated with clemastine (Clem) at indicated doses for 14 days to assess their renewal capacity. *p*-values indicate significance between vehicle and clemastine-treated groups. (**K**) ELDA of BTIC#102 (left panel) and BTIC#148 (right panel) cells with or without 15-day clemastine (4 µM) pre-treatments (pre-Tx) and/or subsequent two-week clemastine (4 µM) treatments (cont. Tx). *p*-values indicate significance between the pre-Tx^−^cont. Tx^−^ group and other groups. *n* = 4 per condition. “-“: no clemastine in the media; “+” with clemastine in the media. Data are represented as (**A**) geometric mean ± geometric S.D. of fold change relative to vehicle-treated groups or (**E**,**F**) mean ± S.E.M. (**E**,**F**) Significance was calculated using unpaired *t* tests (F left panel with Welch’s correction) and represented as * *p* < 0.05, ** *p* < 0.01, *** *p* < 0.001, ns: not significant.

**Figure 3 cancers-15-04619-f003:**
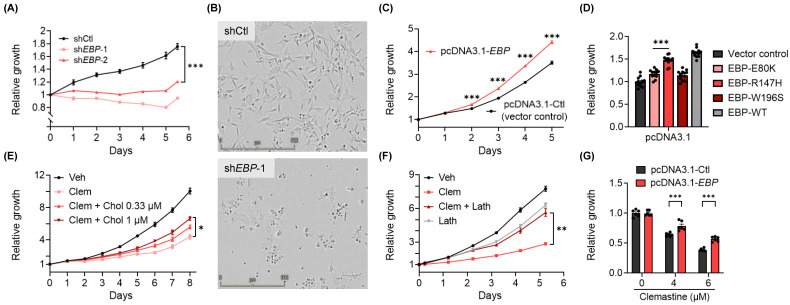
EBP is essential for BTIC proliferation, and the inhibition of EBP contributes to the suppressive effects of clemastine on BTICs. (**A**) Proliferation of non-targeting shRNA control (shCtl) and two non-overlapping *EBP* shRNA (sh*EBP*-1 and sh*EBP*-2) BTIC#102 cells 24 h after lentiviral transduction. *n* = 6 per condition. (**B**) Representative images (4×, scale bar: 500 µm) of shCtl and sh*EBP*-1 BTIC#102 cells 9 days after lentiviral transduction. (**C**) Proliferation of vector control (pcDNA3.1-Ctl) and *EBP*-overexpressed (pcDNA3.1-*EBP*) BTIC#102 cells. *p*-values indicate significance between the two groups at indicated timepoints. *n* = 6 per condition. (**D**) Quantification of relative cell proliferation of BTIC#102 cells with vector control, overexpression of wild-type EBP, or overexpression of each of the three mutant EBPs. The *y*-axis represents normalized phase area confluence at day 6 (normalized to day 0 and then normalized to respective vehicle controls). *n* = 12 per condition. (**E**) Proliferation of BTIC#102 cells treated with clemastine (Clem; at 6 µM) with or without water-soluble cholesterol (Chol) at indicated doses. *n* = 4 per condition. (**F**) Proliferation of BTIC#102 cells treated with clemastine (Clem; at 6 µM) and/or lathosterol (Lath; at 6.25 µM). *n* = 6 per condition. (**G**) Quantification of relative cell proliferation of vector control and *EBP*-overexpressed BTIC#102 cells treated with clemastine at indicated doses. The *y*-axis represents normalized phase area confluence at day 4 (normalized to day 0 and then normalized to respective vehicle controls). *n* = 6 per condition. (**A**, **C**–**G**) Data are represented as mean ± S.E.M. Significance was calculated using two-way repeated measures ANOVA followed by (**A**) Dunnett’s, (**C**) Sidak’s, or (**E**,**F**) Tukey’s multiple comparisons tests, (**D**) ordinary one-way ANOVA followed by Dunnett’s multiple comparisons tests, or (**G**) two-way ANOVA followed by Sidak’s multiple comparisons tests, and was represented as * *p* < 0.05, ** *p* < 0.01, *** *p* < 0.001.

**Figure 4 cancers-15-04619-f004:**
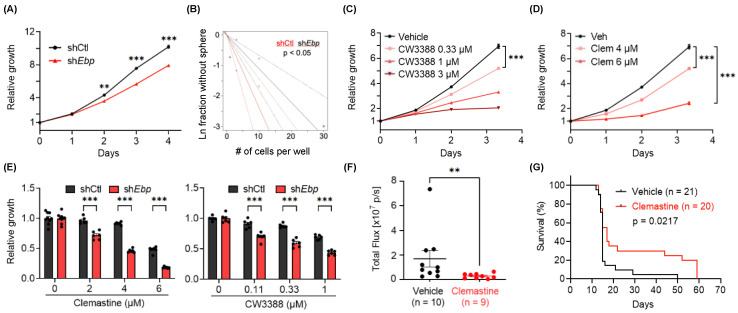
Genetic knockdown and small-molecule inhibition of Ebp suppress the growth of C266 mouse glioma cells in vitro and in vivo. (**A**) Proliferation of non-targeting shRNA control (shCtl) or *Ebp* shRNA (sh*Ebp*) C266 cells 12 days after lentiviral transduction. *p*-values indicate significance between the two groups at indicated timepoints. *n* = 18 per condition. (**B**) ELDA of shCtl and sh*Ebp* C266 cells to assess their self-renewal capacity. (**C**) Proliferation of C266 cells treated with CW3388 at indicated doses. *n* = 6 per condition except vehicle, *n* = 12. (**D**) Proliferation of C266 cells treated with clemastine (Clem) at indicated doses. *n* = 6 per condition except vehicle, *n* = 12. (**E**) Quantification of relative cell proliferation of shCtl and sh*Ebp* C266 cells treated with clemastine (left panel) or CW3388 (right panel) at indicated doses. The *y*-axis represents normalized phase area confluence at day 4 (normalized to day 0 and then normalized to respective vehicle controls). *n* = 6 per condition except for all vehicle groups of the clemastine-treated panel, *n* = 9. (**F**,**G**) In vivo orthotopic mouse models derived from C266 cells treated with vehicle (30% DMSO in PBS) or clemastine (30 mg/kg) five times per week. (**F**) Quantification of tumor sizes 7 days after the first treatment by in vivo bioluminescent imaging and (**G**) Kaplan–Meier analyses. (**G**) Day 0 on the *x*-axis indicates the treatment start date. (**A**, **C**–**F**) Data are represented as mean ± S.E.M. Significance was calculated using two-way repeated measures ANOVA followed by (**A**) Sidak’s or (**C**,**D**) Dunnett’s multiple comparisons tests, (**E**) two-way ANOVA followed by Sidak’s multiple comparisons tests, (**F**) Mann–Whitney tests, and was represented as ** *p* < 0.01, *** *p* < 0.001.

## Data Availability

All codes and data use will be available upon request. The data for RNA-seq were deposited with assigned GEO accession numbers GSE186319 and GSE186392.

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
