# Peer review of "Repurposing Clemastine to Target Glioblastoma Cell Stemness"

_cancers, 2023, doi:10.3390/cancers15184619_

Round 1
Reviewer 1 Report
Authors present an in-vitro study on primary GBM cells to demonstrate that clemastine, an over-the-counter drug for treating hay fever and allergy symptoms, effectively attenuated the stemness and suppressed the propagation of primary brain tumor initiating cells (BTICs) cultures bearing PDGFRA amplification, accompanied by altered gene expression profiling indicative of their more differentiated states, resonating with the activity of clemastine in promoting the differentiation of normal oligodendrocyte progenitor cells (OPCs) into mature oligodendrocytes. Emopamil Binding Protein (EBP), an enzyme in the cholesterol biosynthesis pathway, was found to be essential for BTIC propagation. Introduction provides conclusive information and Materials and methods are well written, Results conclusive. It is however important to broaden the Discussion, since this would be the first publication on use of clemastine for GBM cell cultures. I suggest to include a literature review on OTC medication which has shown effectiveness, and provide a Table. Limitations as well as futur directions should be included.
Acceptable.
Reviewer 2 Report
In this study, the authors investigated the role of clemastine, an over-the-counter anti-histaminic/anti-cholinergic drug commonly used for alleviating allergy symptoms, in attenuating the stemness and propagation of BITC (brain tumor-initiating cells) with a proneural (PN) phenotype (which are characterized by PDGFRAamplification). Among GBM subtypes, the PN tumors are characterized by stem/progenitor-like signatures, which makes them a suitable model for studying the impact of therapeutics on BTIC. Key findings of the study are: (i) clemastine has a tumor differentiation effect on BTIC enriched in stem/progenitor-like transcripts, which may explain the tumor cell growth-inhibitory effect observed with this drug in BTIC models, (ii) functional analyses of clemastine targets in BTIC conducted by the authors further identified the Emopamil Binding Protein (EBP), an enzyme involved in the cholesterol biosynthesis pathway, as an essential molecular player for BTIC propagation and a prime target of clemastine, and (iii) clemastine has a tumor growth-inhibitory effect in vivo when tested in a neural stem cell-derived mouse syngeneic model of GBM.
This is a very well-written manuscript with convincing data, and I congratulate the authors for the scientific rigor by which they conducted their study. The conclusions of the study are generally supported by the presented data, and, for this reason, my general recommendation for the manuscript is to be accepted for publication. However, there are a few points (please see my comments below) that deserve a further clarification in the discussion section of the manuscript.
1. The hypothesis that EBP is essential for the BTIC state is well-worth exploring but was not fully validated by the authors in vivo. Were the shEbp mouse BTIC ever tested in vivo for their ability to form GBM tumors? Unless I am missing something here, limiting dilution experiments conducted with these cells in vivo—which are currently lacking—will further validate the model used by the authors by clearly demonstrating the involvement of this enzyme in the tumor initiation process.
2. What are the clinical implications of using a drug like clemastine in the vision of the authors? In other words, how do the authors think clemastine could be integrated with standard therapy for GBM? Is targeting the BTIC aspect still relevant in patients with newly-diagnosed GBM who already have progressive tumors (which are well past the tumor initiation point) at the time of diagnosis? Furthermore, the ability of the Stupp protocol for newly-diagnosed GBM patients to kill tumor cells is reliant on the capacity of tumor cells to proliferate rapidly. This form of therapy is believed to be self-limiting as a higher proportion of surviving GBM cells become growth-arrested under treatment pressure. Moreover, a high proportion of GBM tumors also shift to a mesenchymal phenotype under the pressure of the Stupp protocol. For this reason, the authors’ thoughts on how to integrate an anti-proliferative drug like clemastine in the current scheme of treatment deserve further discussion.
3. Lastly, it would be important for the authors to further clarify the paradoxical effect of clemastine in shEbp mouse cells. The fact that these cells are still sensitive to clemastine after Ebp knockdown points to a more complex picture that needs further clarification.
